# Ultrasonic Atomization as a Method for Testing Material Properties of Liquid Metals

**DOI:** 10.3390/ma17246109

**Published:** 2024-12-13

**Authors:** Wojciech Presz, Rafał Szostak-Staropiętka, Anna Dziubińska, Katarzyna Kołacz

**Affiliations:** 1Faculty of Mechanical and Industrial Engineering, Warsaw University of Technology, Narbutta 85 Street, 02-524 Warsaw, Poland; anna.dziubinska@pw.edu.pl; 2Lukasiewicz Research Network—Tele and Radio Research Institute, Ratuszowa 11 Street, 03-450 Warsaw, Polandkatarzyna.kolacz@itr.lukasiewicz.gov.pl (K.K.)

**Keywords:** liquid metal, properties testing, viscosity, surface tension, ultrasonic, atomization, simulation

## Abstract

Ultrasonic atomization is an object of steadily increasing interest from metal powder manufacturers, both for additive manufacturing and powder metallurgy. Based on the analysis of available theoretical studies, simulations and experiments, it was noted that the average particle size after atomization and the final particle size distribution depend on the process parameters (e.g., frequency, amplitude) and the parameters of the atomized fluid (e.g., viscosity, surface tension). The objective of this study is to evaluate the feasibility of using ultrasonic atomization to study the properties of liquid metals. It attempts to close a gap in existing knowledge in searching for a new, possibly simple and cost-effective method to study the properties of liquid metals and clarify the relationship between ultrasonic atomization parameters (amplitude, frequency, metal spill on vibrating surface) and obtained atomization results (average particle size, particle size distribution, atomization time). Utilizing numerical modeling as a methodology, especially the finite element method, the possibilities of using ultrasonic atomization as an instrument to determine properties of liquid metals were considered as an introduction to a series of real experiments. Modeling was applied to liquids with different properties, atomized at a chosen specific constant frequency and amplitude. The results of the simulation are in line with the current state of knowledge about ultrasonic atomization. However, in the existing studies available to the authors, there are no data that can be compared directly, but indirect comparisons confirmed the conclusions of the preliminary literature analysis. The relationship between viscosity and surface tension and the average size of the atomization processes obtained in the simulation of particles was demonstrated, thus providing a tool for the development of the presented concept: ultrasonic atomization as a research method. Research and simulation results led to the final conclusion: ultrasonic atomization can be applied to study the properties of liquid metals and this will be the subject of further research and experimentation.

## 1. Introduction

Material properties such as viscosity and surface tension are important parameters influencing the behavior of liquid metals in various technological processes. In metallurgical and foundry processes, these parameters have influence, e.g., on kinetics of the reaction, which is important from the point of view of the following processes: metal refining, mold pouring, process of degassing the casting in the solidification process, as well as the metal solidification process in the mold itself.

Primary crystallization is an important factor because it creates a structure that determines many properties of metals and their alloys. The mechanical and plastic properties of the alloy depend on the primary structure of the casting or ingot, which thus determines the technology of further processing of raw materials and the possibility of various applications of these materials [1].

The process of testing the properties of liquid metals is also difficult due to the difficulty of keeping the liquid metal clean due to the contamination of metals with surfactants. Knowledge of the technological properties of liquid metals allows assessment of the quality of liquid metal, diagnosis of problems and facilitation of the design of further machining processes. For this reason, many of the methods commonly used to determine the properties of liquids are not applicable to the study of the properties of liquid metals.

The method proposed in this article is a method in the initial phase and at this stage it does not take into account contaminations, i.e., the condition and amount of slag in the steel, but there are strong reasons to also use this method for testing slag condition or metal/alloy purity. At this moment, these simulations were performed only for metals of high purity and homogeneity.

The most important parameters characterizing fluids, and thus metals in a liquid state, are viscosity and surface tension. Viscosity is defined as the internal friction in a fluid and described as the ability to carry tangential stresses by the fluid. Surface tension, on the other hand, is defined as a physical phenomenon that occurs at the interface between the surface of a liquid and another medium, thanks to which this surface behaves like an elastic film. The cause of this phenomenon is believed to be the occurrence of attractive forces between the molecules of the liquid. This article is an attempt to find the answer to the question whether ultrasonic atomization can be an effective tool to study the properties of liquid metals, including impurities. This will be examined later, in detail, during real experiments.

## 2. Common Methods of Testing Properties of Liquid Metals

With the development of machining processes, methods for testing metals in the liquid state have developed and still continue to develop. The common methods of testing the viscosity of metals in the liquid state summarized in [2,3] include: rotational viscometer, capillary method, vibrating spindle technique, gas bubble viscometer, falling object method, draining vessel and measuring the damping of acoustic wave. Recently, in relation to the study of liquid metal impurities, the method of measurement by means of gas atomization has also been described [4].

The most commonly described and used methods will be discussed below.

### 2.1. Rotational Viscometer Method

This method is based on measuring the resistance forces of an object immersed and rotating in a vessel filled with liquid metal. Its advantage is the ability to perform measurements at different rotation speeds and thus at different values of shear forces. The method works well in the measurement of liquids with low viscosity, provides easy control of the measurement temperature, avoids errors caused by the access of the atmosphere (the rotating measuring system is constantly immersed in the liquid) and it is also possible to carry out measurements in the atmosphere of any gas due to the possibility of covering the measuring system and isolating the liquid from the atmospheric air.

However, the measuring system must be very precisely axially aligned to eliminate measurement interferences caused by runout. This is especially important when measuring at high speeds and high temperatures. Measurements at high rotational speeds can also be affected by turbulent flow and other fluid instabilities. In measurements carried out by this method, the factors of the size of the rotating element itself, its shape and its distance from both the walls of the vessel and from the bottom as well as the surface of the liquid in the vessel are important. The viscosity is determined as a function of the measured torque, rotational speed and radius of the submerged object. The method for determining viscosity is mentioned in [2,3] and detailed research is described in [5].

### 2.2. Capillary Method

This method is based on measuring the time in which a certain amount of pressurized fluid flows through a vertical hole with a diameter of 0.15–0.2 mm and a length of 70–80 mm [2,6]. An important factor influencing the quality of the measurements is the need to ensure laminar flow. In addition, it is important to ensure the homogeneity of the tested fluid, the presence of gas bubbles as well as, e.g., oxides or other inclusions which have a negative impact on measurements, including flow blockage [2,7].

The method is practically used for temperatures of 1200 °C. It is described as a comparative method, and the results of measurements are related to other measurements and dependent on the geometry of the hole, the height of the liquid column above the hole and the density of the tested fluid. Modification of the method described in [7] also makes it possible to test the surface tension.

### 2.3. Oscillatory Viscometer

In this test, a cylindrical vessel containing a liquid is set in an oscillatory motion relative to the axis of a vertically set vessel (cylinder). This motion, after the excitation force ceases, is dampened by friction and the dissipation of energy in the fluid. The viscosity is determined by measuring the changes in vessel motion (amplitude and oscillation period). The method allows for relatively easy measurement of both the period and amplitude of motion as well as maintaining a constant temperature during the test [8].

Although this method is the most common due to its simplicity and relatively accurate measurement results, it is dependent on relating the results of measurements of vessel motion to the viscosity itself by the quadratic equation of oscillatory motion. For this reason, viscosity measurements often have different results for similar conditions, which necessitates analytical corrections [8,9,10,11,12]. A schematic diagram of an oscillatory viscometer for liquid metals is shown in Figure 1.

### 2.4. Falling Object Method

This method measures the speed or time of descent for a body (typically spherical in shape) in a vessel with liquid metal under the influence of gravity force [13]. However, it is necessary to correct the dimensions of the spherical mass due to the change in diameter caused by its thermal expansion. It is also an indirect method that requires calibrating the measuring system to liquids of known viscosity. For the measurement results, it is advantageous that the diameter of the submerged object is not greater than 1/10 of the diameter of the vessel [2].

### 2.5. Draining Vessel Method

This method is used to make comparative viscosity measurements of different liquids and involves measuring the time it takes for a certain amount of liquid to flow freely out of a vessel. Although it has the advantage of simplicity, the measurement results depend on the geometry of the vessel and changes in the pressure of the fluid column to the bottom associated with the decrease in the fluid level. Other limitations are similar to those of the capillary method, however, the method has been successfully used to determine the viscosity, surface tension and density of liquid metal [14]. Due to the specificity of liquid aluminum, the results of measurements of its alloys give underestimated values.

On the basis of the conducted tests and measurements, many mathematical models have been developed to determine viscosity, among which the most commonly used are the equations of Andrade, Arrhenius and Kaptay described and summarized in [2,3]. On the basis of measurements and mathematical models as well as the results of measurements of pure metals, many models have also been developed to determine the viscosity of the alloys mentioned in the study [3]. The generalized model by Budai et al. [9] effectively predicts alloy viscosities even with unknown viscosity of individual components but struggles with incongruent melting alloys [3].

Summarizing, the above-mentioned methods are based mainly on the study of the resistance of movement of a body immersed in liquid metal or of a movement of liquid metal in relation to stationary walls. Each method requires complex equipment and a relatively large amount of liquid metal.

## 3. The Method of Gas Atomization as an Attempt to Determine the Material Properties of a Liquid Metal

Recently, due to the development of powder metallurgy and the development of atomization techniques for liquid metals, studies of the properties of liquid metals have also been carried out using these methods. The experiment was carried out using the inert gas atomization method. Process control is provided by technological parameters (mainly hardware) [4,15,16] such as temperature, pressure and flow rate of gas, as well as the flow rate of the liquid metal.

However, the authors of [4] note that a change in the particle size distribution (PSD) is possible. Moreover, it depends on the properties of the liquid metal, such as surface tension, viscosity and density, and assuming that these properties are temperature-dependent, it is possible to also control the particle size by changing the temperature.

According to the Lubanska equation [16,17], the average particle size in the gas atomization process in relation to the diameter of the stream of liquid metal supplied to the chamber is given as follows:(1)$$\frac{d_{m}}{d_{Met\;Str}}={K\left[\frac{v_{m}}{v_{g}}\frac{1}{W}\left(1+\frac{M}{A}\right)\right]}^{\frac{1}{2}}$$
where: d_m_ is the average particle size (e.g., d_50_), d_Met Str_ is the diameter of the liquid metal stream, ν_m_ is the kinematic viscosity of the liquid metal, ν_g_ is the kinematic viscosity of the atomizing gas, M is the mass flow rate of the liquid metal, ρ is the density of the liquid metal, A is the flow rate of the atomizing gas, K is a process constant depending on the geometry of the atomizing nozzle [16].

Weber’s number (W), that also appears in the equation, is characterized as:(2)$$W=\rho \frac{V^{2}D}{\sigma }$$
where: ρ is the density, σ is the surface tension, V is the velocity of flow, D is the characteristic dimension of the flow (in this case, the diameter of the liquid metal nozzle).

The Weber number itself is defined as the ratio of the inertia forces to the surface tension forces of a liquid and is most often used to determine the impact energy of a liquid on a surface or to calculate the movement of a portion of one fluid in another fluid.

The first conclusion from the analysis of the equation is that the density and viscosity of the liquid contribute to the change in the nature of the flow, and the second is that the reduction of the average particle size can be achieved by increasing the Weber number and thus by influencing the viscosity or surface tension of the liquid metal. Finally, assuming atmospheres of vacuum or various inert gases during the test, the viscosity and the surface tension can be determined on the basis of the obtained particle size. This leads to the assumption that parameters of a process are dependent on the type of gas breaking the metal and thus its effect on the viscosity of the liquid metal should also be taken into consideration.

In studies described in [18,19], the influence of the type of gas on the results of the process, i.e., on the particle size, has been confirmed. According to both studies, an additional factor responsible for the change in the viscosity of the liquid metal is the formation of oxides on the surface of atomized droplets.

The results of tests described in [20,21], carried out on liquid media atomized by a gas stream, show that there is a relationship between the size of the atomized particle and the surface tension and density of the atomized fluid, and it is directly proportional: the larger the particles are, the higher the density, viscosity and surface tension of the fluid.

In [4] the authors did not obtain confirmation of these assumptions, concluding the study with the claim that the change in the ratio of the flow rate of the atomizing gas to the intensity of the liquid metal did not cause a significant change in the size of the atomized particles. At the same time, the opposite conclusion has been presented with regard to changes in surface viscosity and surface tension: increases in surface tension and viscosity cause an increase in the size of the atomized particle. Another conclusion from the experiment is that the simultaneous change in viscosity and surface tension does not cause an additional increase in the particle dimension growth over the changes caused by separate changes in these parameters, and the cumulative effect did not occur.

Despite the relative simplicity and ease of the process [15], certain limitations such as the formation of a solidified metal deposit at the nozzle outlet [22] or even clogging of the liquid metal nozzle [23] are factors that have a proven effect on the flow of both gas and liquid metal and thus on the atomization process. This has a significant impact on the final shape and distribution of the particles and thus on the results of this atomization conducted as a research process.

At this point it is worth mentioning that there are methods similar to gas atomization, i.e., water and oil atomization. Although they are based on the same mechanism of breaking a stream of liquid medium (metal) with another medium and some similarities can be assumed as per Equation (1), there are no presented experimental results that can be related to parameters of this atomization process.

## 4. Considerations on Ultrasonic Atomization as a Method to Investigate the Properties of Liquid Metals

The production of metal powders by ultrasonic atomization has recently become an object of interest for manufacturers due to the many advantages of the process, such as its predictability of average particle size and the possibility of obtaining a narrow distribution of powder particles in relation to other atomization techniques [24]. An advantage of the process is also the high contribution of particles with the desired spherical shape in the total volume of the obtained powder. Other advantages, such as the possibility of carrying out the process in an atmosphere of different gases (inert and active) under both vacuum and pressure and the relatively low velocities of the atomized particles affecting the overall size of the atomization device, have led to interest in the ultrasonic atomization method not only for purely commercial purposes but also for research purposes. The cost of the process is also worth mentioning here, as it can be significantly reduced by the possibility of elimination of the expensive inert gases from the process.

The process of ultrasonic atomization of liquids as a derivative of standing wave generation has been known for more than 50 years and is quite well described in the literature [25,26]. It is most often depicted as the detachment of liquid droplets from the wave-tops caused by vibrations in the direction of the thickness of the liquid layer [24,25,26,27,28], although [29] describes the possibility of inducing atomization only through the action of a standing wave caused by changes in the pressure of the medium in which the liquid metal stream is located.

Waves on the surface of a fluid caused by ultrasound propagating perpendicular to the surface have been the subject of many studies: e.g., [30] describes the formation of waves by ignoring viscosity and assuming a small amplitude on the surface causing vibrations. In [31], it is shown that the frequency of capillary waves in ultrasonic atomization is equal to half of the forcing frequency of the source causing the vibrations. On the basis of the measured capillary wavelength and the particle size excited in the ultrasonic atomization process according to [32], the author indicates the relationship between the average particle diameter (d) and the capillary wavelength (λ) as:*d* = 0.34 · λ(3)

The above equation refers to the capillary wavelength [25,33] and the results of the tests [31], the relationship between the size of the atomized particle and the parameters of the process as well as the atomized liquid [32] can be clarified as:(4)$$d={0.34\;\left(\frac{8\;\pi \;\sigma }{\rho \;f^{2}}\right)}^{\frac{1}{3}}$$
where ρ and σ are respectively the density and surface tension of the atomized fluid and f is a vibration frequency. This equation has proven its practical significance in many observations and experiments [24,28]. It gives a good initial estimation, however, recent studies show that the size of the atomized particle is related to liquid parameters with a constant different from 0.34 and within a wider range, dependent on ultrasonic devices.

As per the formula above, surface tension can be found as:(5)$$\sigma =\frac{\rho \;f^{2}\;d^{3}}{0.314432\;\pi }$$

A number of theoretical analyses regarding this behavior [34,35,36,37] can be found in the literature. They assume a sufficiently thin layer and excitation in a direction perpendicular to the surface of the fluid. One of the practically realized experiments aimed at linking existing dependencies and concluding on the scalability of the derived formulas is, e.g., the experiment described in [33]. One of the conclusions of the experiment is the determination of the critical amplitude, necessary to start the process of wave excitation and thus atomization:(6)$$A_{m}=\frac{2\mu }{\rho }\;\cdot \;\sqrt[{3}]{\frac{\rho }{\pi \;\sigma \;f}}$$
where A_m_ is the critical amplitude, μ is the viscosity, ρ is the density, σ is the surface tension of the liquid and f is the frequency.

The second conclusion is on connecting the relationship between flow quantities such as the Weber number and the Reynolds number with the quantities characteristic of the atomization process, such as frequency and amplitude. Examples of practical, more or less accurate implementations of ultrasonic atomization are described in [24,28,38] and confirm the applicability of the model [32] under real conditions, with accuracy regarding the assumption of constant values of flow numbers (Weber, Ohensorge and intensity number).

The relationship between the size of the atomized particle and the parameters of the process can then be described as follows [39]:(7)$$d={\left(\frac{\pi \sigma }{{\rho f}^{2}}\right)}^{0.33}\left[1+{A(We)}^{0.22}+{\left(Oh\right)}^{0.166}+{I_{N}}^{-0.0277}\right]$$
where the Weber number is modified to include flow rate (Q) and ultrasonic frequency (f) as follows:(8)$$W=\frac{fQ\rho }{\sigma }$$

The Ohnesorge number (known also as viscous number) is modified to include amplitude (A_m_) and is defined as follows:(9)$$O_{h}=\frac{\nu }{f\;{A_{m}}^{2}\rho }$$

The intensity number that takes into account the effect of energy density on droplet size is defined as:(10)$$I_{N}=\frac{f^{2}{A_{m}}^{4}}{c\;Q}$$
where c is the velocity of sound in a liquid medium.

The right-hand side of Equation (7) [39] indicates the influence of parameters other than surface tension and frequency included in Equation (4).

In studies related to ultrasonic atomization, the Weber number can refer to the diameter of the atomized droplet as the previously mentioned characteristic dimension, to the flow rate of the liquid to be atomized [39] or to the thickness of the atomized liquid layer [33].

The research presented below is an attempt to confirm the scale of the relationship between the material properties characterized by the Weber number and the parameters of the atomization process, assuming an approximately constant density of the atomized liquid.

## 5. Assumptions for Planned Experiment

The starting point for the planned experiment of liquid metal atomization is the numerical verification of the hypothesis on the dependence of the size of the atomized particle on the properties of the liquid metal. Even the above-mentioned literature examples are limited to identifying empirical relationships and predicting certain results on their basis, rather than presenting the results of simulations or research. This still makes it difficult to predict the size and distribution of atomized particles with sufficient accuracy [40,41]. An interesting attempt to relate the frequency of excitation and particle size is a series of numerical simulations described in [42]. It is based on the numerical volume of fluid (VOF) method as one of the available methods for modeling multiphase flows using computer-aided design methods in the field of computational fluid dynamics (CFD).

VOF is a multiphase model that is considered to be simple but predictable and highly accurate [43]. Its advantage is that it uses only a single momentum equation [42]:(11)$$\frac{\partial }{\partial t}\left(\rho \;\vec{v}\right)+\nabla .\left(\rho \;\vec{v}\vec{v}\right)=-\nabla p+\nabla .\left[\mu \left(\nabla \;\vec{v}+\nabla \;{\vec{v}}^{T}\right)\right]+\rho \vec{g}+\vec{F}$$
where contribution of the fractions depends on density and viscosity.

The vibrating motion of the base causing vibrations in the liquid is inflicted according to the continuity equation:(12)$$\frac{1}{\rho_{y}}\left[\frac{\partial }{\partial t}\left(\alpha_{y}\rho_{y}\right)+\nabla .\left(\alpha_{y}\rho_{y}{\vec{v}}_{y}\right)=S_{\alpha y}+\sum_{x=1}^{n}\left({\dot{m}}_{xy}-{\dot{m}}_{yx}\right)\right]$$
where m_yx_ and m_xy_ denote the mass exchange between the x and y as well as y and x phases, respectively, S_αy_ as the mass source for each phase is a constant value in this case. The surface tension is a constant value corresponding to the value for the liquid to be tested. For the standard interpolation scheme, the values of each fraction are determined from the following equation:(13)$$\frac{\alpha_{y}^{n+1}\rho_{y}^{n+1}-\alpha_{y}^{n}\rho_{y}^{n}}{∆t}+\sum_{f}\left(\rho_{y}U_{f}^{n}\alpha_{y,f}^{n}\right)=\left[S_{\alpha y}+\sum_{x=1}^{n}\left({\dot{m}}_{xy}-{\dot{m}}_{yx}\right)\right]V$$
where n + 1 is the index for the current time step, n is the index for the previous time step, α_y,f_ is the factor of filling the element with the y fraction, V is the volume of the element, U_f_ is the volume flow through the element.

## 6. Numerical Model of Ultrasonic Atomization

First, the theoretical considerations about ultrasonic atomization were verified against a mathematical model. In order to verify the correctness of the assumptions, a model similar to the one described in [42] was constructed. In order to shorten the calculation time, modeling space was reduced to the dimensions of 0.002 m (2 mm) × 0.0004 m (0.4 mm). The thickness of the liquid layer on the vibrating surface is 0.00004 m (0.04 mm). These similarities were made to be consistent with the model mentioned above in order to significantly ease verification of simulation results. Calculations were performed using the Fluent solver included in the ANSYS R19 package [44]. A diagram of the calculation model is shown in Figure 2.

To minimize the influence of the mesh on the calculations, a homogeneous quadrilateral mesh with a size of 0.002 mm was used. The model constructed on the basis of these criteria consists of 200 000 elements. In the first stage, like in the simulation described in [42], water with the following parameters was used as a liquid to check the adopted simplifications in the model:

density: 998 [kg/m^3^];

viscosity: 1.003 [mPa·s];

surface tension: 0.0728 [N/m].

The model is based on the equation of motion of the ground defined as:y(t) = A·sin(2π·f·t)(14)
where y(t) is the deflection as a function of time, A is the amplitude, f is the frequency, t is time.

In order to indirectly verify the model and results with the results published so far, i.e., in [24,28,42], the frequency of f = 50 kHz was used in the simulation as a reference to the results presented in the above-mentioned studies.

## 7. Simulations, Results and Comments

In the first phase, vibrations with an amplitude of 0.02 mm, or 20 μm, were simulated to provide conditions for wave excitation, in terms of energy able to overcome viscosity and surface tension forces. The first simulation indirectly confirmed the assumptions and results [42], but according to the adopted model, the layer of water atomized completely in 0.000922 [s]. On the other hand, the mechanism of wave excitation and the dependence of its length on frequency, determined at the level of 0.1162 [mm] and visible in Figure 3, were confirmed. On the basis of this value, the average diameter of the particle, determined from the theoretical formula, is 39.5 μm.

According to the simulation, the size of a single drop of liquid, resulting from atomization, detached from the wave tip was determined as 36.3 μm, which confirms both the results of the experiments described in [24,28] as well as the common Formula (3) for the size of the particle. Accuracy of the result is approx. 3 μm, determined on the basis of the size of the adopted mesh element. The process of droplet formation and their further evolution over time is shown in Figure 4.

A natural consequence was an attempt to determine the minimum vibrational amplitude necessary for the atomization process. For this purpose, another simulation was carried out, this time for an amplitude half as small, i.e., 10 μm. The results of atomization after a time of 0.0119 [s] (about 10 times longer) are shown in Figure 5.

Observation of the results and the course of the simulation allows us to present the thesis that the amplitude of vibrations at the level of 10 μm is too small to induce atomization of liquids with parameters similar to water when the layer of liquid has the thickness mentioned earlier. After a time of 0.0119 [s], the liquid layer does not atomize, it only agglomerates into larger droplets with a thickness of approx. 0.087 [mm] (87 μm). On their surface, waves of a determined length of approx. 0.1125 [mm] are formed, coinciding with the wavelength excited on a uniform layer. Droplets do not tend to spill fully over the vibrating surface.

The result of this simulation mostly confirms the thesis from studies [38,39] according to which atomization will not occur if the amplitude of vibrations is smaller than the critical amplitude. For the atomization of liquid aluminum, this value was determined to be 40 μm. In the next step, in order to verify the hypothesis linking droplet size with viscosity and surface tension, simulations were carried out for three different liquids (liquid metals) with the following parameters. For comparison, water is also included in the table (Table 1):

In the next stage of considerations, the thickness of the liquid layer in the model was increased from 0.00004 m (0.04 mm) to 0.00005 m (0.05 mm) and 0.00006 m (0.06 mm). The purpose of the change was to check the influence of the thickness of the liquid layer on the parameters of the atomization process. No significant changes were observed in the time of wave formation and the formation of the first droplets when the layer thickness increased from 0.04 mm to 0.05 mm, however the time of complete atomization of the liquid layer with a further 25% increase in thickness (from 0.04 to 0.06 mm) was increased by more than 80%.

## 8. Conclusions

The aim of the study was to evaluate the feasibility of using ultrasonic atomization to study the properties of liquid metals and the possibility to use numerical simulations as approximations of real experiments. The relative convergence of the simulation results with the available data from other experiments [24,28,45] confirms that the correctness and accuracy of the model are sufficient to make a decision about the construction of a test stand and the execution of real experiments. The presented results are a subject for further discussions, as results available in the literature were obtained from experiments conducted with not exactly the same input and process parameters. The importance of parameters is emphasized more and more often due to the large discrepancies in the obtained results. According to [46], the coefficient connecting the diameter of the obtained mean droplet with the wavelength, according to Equation (3), may vary in the range of 0.17 to 0.65. There is still a lack of experimental results that can be compared both directly and indirectly. In fact, only the authors of [39] performed experiments showing an effect of the dynamic viscosity but considered viscosity greater than 10 cP. These experiments, however, were not performed on liquid metals.

One of the results presented in this article, which allows for some inferences and further research on the process of atomization of liquid metals, is undoubtedly the observed process of the formation of only larger agglomerates (droplets) of liquids instead of proper atomization at an insufficient amplitude value. The conclusion is that the theoretical predictions of the excitation of the capillary wave are correct for the excitation of the wave itself but are not entirely correct for the process of initiation of atomization, understood as the detachment of a liquid droplet from the top of the resulting wave. The results of the research presented in [45] partially confirm the previous theoretical considerations in the field of the dependence of atomization results on frequency and amplitude in the range of approx. 7.5 to 38 kHz and partially contradict them. According to this study, the amplitude necessary for the process of initiating atomization is about 60–70% higher than theoretically predicted. It has been confirmed that, as the frequency increases to about 22 kHz, the amplitude necessary to start the atomization process decreases to about 2.0 × 10^−6^ m (2 μm) and then increases linearly to about 3.0 × 10^−6^ m (3 μm). In the experiment, the atomized liquid was water. One of the conclusions of the study [45] is the need for further research due to the lack of experimental results and the development or correction of analytical methods. The study does not provide details on the geometry of the ultrasonic tools used, such as the size of the atomizing surface and thus the size of the liquid droplet, or indirectly the thickness of its layer. Although tendencies in the dependence of the wavelength on frequency observed in the experiment are largely consistent with theoretical considerations, they do not fully support the theory. This means that liquid properties such as viscosity and surface tension may have a larger impact on the atomization process than was originally assumed. Also, the processes of formation of a single droplet observed in a number of simulations carried out for the purposes of this study are not the same for each droplet.

Further study on the importance of process parameters in ultrasonic atomization seems to be justified for one more reason: parameters and liquid properties correlated in the wrong way can even stop the process of ultrasonic atomization, as mentioned in [47]. Excessive feeding resulted in the formation of a considerable amount of molten metal on the sonotrode platform. This phenomenon prevented the ultrasound system from working properly. Nevertheless, some observations and results were provided for further discussion. They cannot be directly compared to the results of this study, as assumptions for the experiment as well as process parameters were different. The amplitude of vibrations given in [%] instead of units of length (i.e., micrometers) makes the results valid only for specific configuration of vibrating stack.

According to the results of the presented simulation experiments, it can be concluded that in some range they are dependent on the thickness of the liquid metal layer, the shape of its surface and the cohesive forces in the liquid metal itself. These factors mean that the droplets do not always atomize in the direction perpendicular to the vibrating surface and to the surface of the liquid itself; their trajectory is changed by the forces of coherence. The result of this disturbance is, for example, the collision and merging of smaller droplets into larger ones or the formation of so-called “satellites” or, using the terminology of powder metallurgy, “grains of undesirable shape”, in which smaller particles stick to larger ones but do not fully combine with them. The mechanism of uneven wave excitation is explained in [33], and the results of this study’s simulations confirm this assumption.

Having in mind the above-mentioned aspects of the atomization process, despite some ambiguities, it is possible to confirm the thesis given at the beginning that ultrasonic atomization can be an effective tool to study the properties of liquid metals, including impurities, however, these cases will be examined later, in detail, during real experiments.

## 9. Future Work

In the next stage of work, a test stand with the ability to atomize liquids with temperatures of over 700 °C, that is, water and some liquid metals, will be constructed. This is the temperature sufficient to melt alloys of aluminum, a metal sensitive to oxidation and thus to changes in viscosity and surface tension. In parallel with the construction of the test stand, a number of more detailed simulations will be carried out as an attempt to determine the influence of the shape of the atomizing surface and the way the droplets spread over the surface during the atomization process. Research is planned as the following steps:

a. Numerical simulation of the atomization process at the given parameters: frequency, amplitude, thickness of the atomized liquid layer, properties of the atomized material.

b. Theoretical determination of the minimum amplitude and the influence of the shape and thickness of the atomized liquid layer on the time of the atomization process.

c. Verification of theoretical analyses and conducting atomization experiments for specific process parameters (specific frequency, specific amplitude).

d. Investigation (separately) of the influence of the amplitude, frequency, shape and thickness of the liquid metal layer and the parameters of the atomized liquid (density, viscosity, surface tension, purity) on the time of the atomization process and the size of particles.

The results will be used for further work on the development of research methodologies for ultrasonic atomization in testing properties of liquids and ultrasonic systems for use in research work on the atomization of liquids including liquid metals and the powder metallurgy industry. After development of methods and equipment, industrial research on powder metallurgy, including research on impurities and slag condition, is also planned.

## Figures and Tables

**Figure 1 materials-17-06109-f001:**
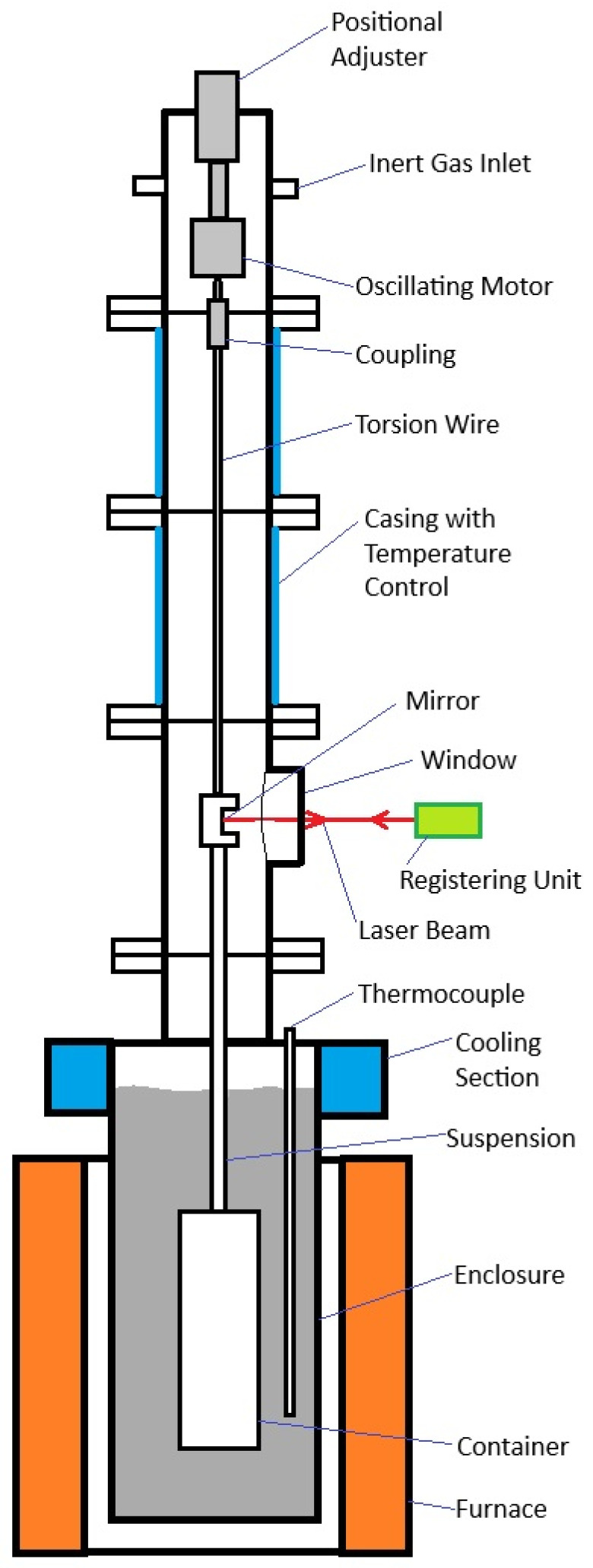
Example scheme of oscillatory viscometer as a precise and complicated equipment.

**Figure 2 materials-17-06109-f002:**
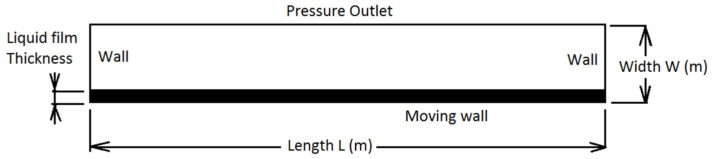
Ultrasonic atomization domain model.

**Figure 3 materials-17-06109-f003:**
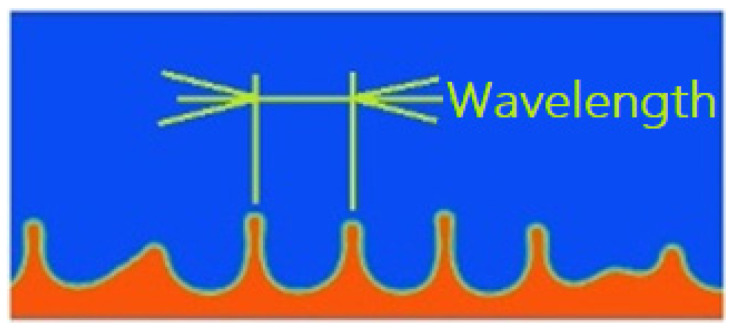
An observed capillary wave of a certain length; air—blue; liquid—red.

**Figure 4 materials-17-06109-f004:**
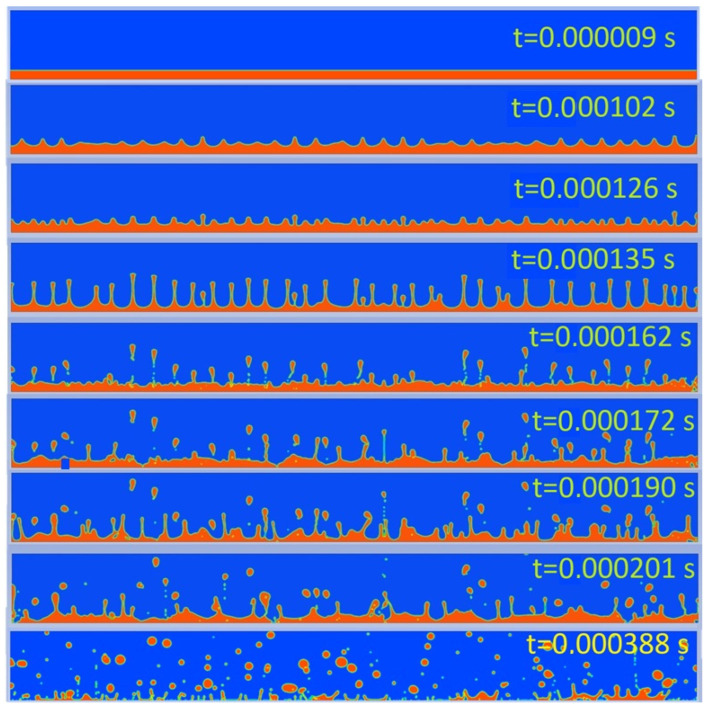
The process of liquid droplet formation and their evolution during ultrasonic atomization; air—blue; liquid—red.

**Figure 5 materials-17-06109-f005:**
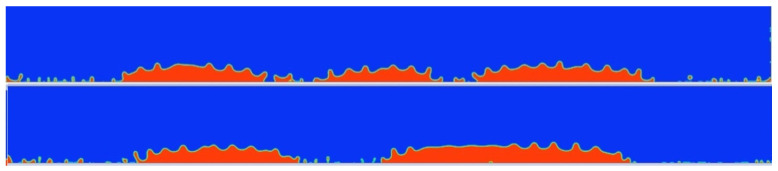
Simulation of liquids’ behavior with too low excitation amplitude; air—blue; liquid—red.

**Table 1 materials-17-06109-t001:** Properties of liquids used in the simulation.

Liquid Medium	Density ρ [kg/m^3^]	Surface Tension σ [N/m]	Dynamic Viscosity μ [mPa·s]	Wavelength [mm]	Droplet Diameter [μm]
Water	998	0.0728	1.003	0.1162	36.3
Aluminum	2630	0.889	1.050	0.1689	49.6
Zinc	6580	0.755	3.500	0.1118	34.8
Steel	7500	1.87	5.300	0.1565	52.1

## Data Availability

The original contributions presented in this study are included in the article. Further inquiries can be directed to the corresponding author.

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
