# Peer review of "Ultrasonic Atomization as a Method for Testing Material Properties of Liquid Metals"

_materials, 2024, doi:10.3390/ma17246109_

Round 1
Reviewer 1 Report
Comments and Suggestions for Authors
Dear authors,
I would like to thank you for the opportunity to review the article. The research presents a good scientific contribution related to the atomization of liquid metal particles. However, adjustments are necessary for the publication of the article.
The following points need to be adjusted:
1. In the abstract, the knowledge gaps that the article intends to address are not clear. It is necessary that the objectives be indicated, as well as the steps of the research process, such as methodology, results, and conclusions, be briefly addressed.
2. In the introduction, the article presents a general approach under the aspects of reaction kinetics, viscosity, and surface tension. However, in a general approach in steelmaking, the condition of the slag, whether acidic or basic, the activities, fugacity, type of deoxidizers, among others, are fundamental in a macro analysis. It is recommended that only the metallurgy specifically used in the distributor used before atomization be addressed. At the end of the introduction, it is recommended to clearly state the objectives or gaps to which the research seeks answers, and these should be answered in the conclusions.
3. The methodology contains information that can be used in the introduction. It is recommended that a flowchart be included to better understand the tests to be performed. The identification of different methods of testing liquid metal can be used in the introduction and not as a methodology. If it is only a review, it should be indicated in the abstract and title. Figure one should be better explained. The study was based on ferrous or non-ferrous alloys. Please clarify the focus of the material to be simulated, since the metallurgy involved has different nuances in each situation. Why does the article focus on gas atomization and not on water atomization?
4. The results require further discussion with other authors and correlations of the simulation using ultrasound atomization with other techniques.
5. There are results in the conclusions. Ideally, the conclusions should respond to the knowledge areas identified at the end of the introduction.
6. The references should be updated and have a higher impact factor.
Sincerely
Author Response
Dear authors,
I would like to thank you for the opportunity to review the article. The research presents a good scientific contribution related to the atomization of liquid metal particles. However, adjustments are necessary for the publication of the article.
First of all, we would like to thank you very much for your valuable comments on the article, they will help us in our future work. Below are our responses on specific comments.
The following points need to be adjusted:
In the abstract, the knowledge gaps that the article intends to address are not clear. It is necessary that the objectives be indicated, as well as the steps of the research process, such as methodology, results, and conclusions, be briefly addressed.
The information mentioned above is better highlighted in the introduction. It is not expanded much as not to excessively increase the content of the article, but below we refer in detail to each element:
Objective: To evaluate the feasibility of using ultrasonic atomization to study the properties of liquid metals.
Knowledge gaps: Searching for a new, possibly simple and cost-effective method to study the properties of liquid metals and clarifying the relationship between ultrasonic atomization parameters (amplitude, frequency, metal spill on vibrating surface) and obtained atomization results (average particle size, particle size distribution, atomization time)
Methodology: numerical simulation of the process as an introduction to a series of experiments.
Results: The simulation results are in line with the current state of knowledge about ultrasonic atomization. In the existing studies available to the authors, there is no data that can be compared directly.
Conclusion: Ultrasonic atomization can be applied to study the properties of liquid metals and this will be the subject of further research and experimentation.
In the introduction, the article presents a general approach under the aspects of reaction kinetics, viscosity, and surface tension. However, in a general approach in steelmaking, the condition of the slag, whether acidic or basic, the activities, fugacity, type of deoxidizers, among others, are fundamental in a macro analysis. It is recommended that only the metallurgy specifically used in the distributor used before atomization be addressed. At the end of the introduction, it is recommended to clearly state the objectives or gaps to which the research seeks answers, and these should be answered in the conclusions.
The proposed method is a method in the initial phase and at this stage it does not take into account, i.e the condition and amount of slag in the steel, but there are strong reasons to use this method also for testing slag or metal/alloy purity. At this moment these simulations were only for metals of high purity and homogeneity. Gaps: Finding the answer to the question whether ultrasonic atomization can be an effective tool to study the properties of liquid metals, including impurities. This will be examined later, in details.
The methodology contains information that can be used in the introduction. It is recommended that a flowchart be included to better understand the tests to be performed. The identification of different methods of testing liquid metal can be used in the introduction and not as a methodology. If it is only a review, it should be indicated in the abstract and title. Figure one should be better explained. The study was based on ferrous or non-ferrous alloys. Please clarify the focus of the material to be simulated, since the metallurgy involved has different nuances in each situation. Why does the article focus on gas atomization and not on water atomization?
The article mentions gas atomization as one of the methods of atomization, the only one that somehow related the results of the atomization process to the properties of atomized metals. For atomization with the use of other media (water, oil) there are no such studies. Water Atomization is a method very similar to atomization with gas, it is based on the same mechanisms as breaking a stream of liquid medium (metal) with another medium.
The figure caption has been supplemented with the phrase "an example of a complex and precise apparatus for measuring viscosity".
"Method flowchart" was added to in the "future work" paragraph:
- Computer simulation of the atomization process at the given parameters: frequency, amplitude, thickness of the atomized liquid layer, properties of the atomized material.
- Theoretical determination of the minimum amplitude and the influence of the shape and thickness of the atomized liquid layer on the time of the atomization process.
- Conducting atomization experiments for specific process parameters (specific frequency, specific amplitude).
- Investigation of the influence (separately) of the amplitude, frequency, shape and thickness of the liquid metal layer) and the parameters of the atomized liquid (density, viscosity, surface tension, purity) on the time of the atomization process and the size of particles.
- The results require further discussion with other authors and correlations of the simulation using ultrasound atomization with other techniques.
This statement has been added to the body of the article. Refeences to similar results [24] and [28] are added with the note that this is a slightly different method of atomization.
- There are results in the conclusions. Ideally, the conclusions should respond to the knowledge areas identified at the end of the introduction.
A paragraph has been added to the text. The results of the simulation described in the article refer to one specific case of atomization, at a specific frequency and amplitude. These results are consistent with the available theoretical results and results described in [24] and [28]. There are no other published results of the atomization process with parameters as in the article that could be directly compared.
- The references should be updated and have a higher impact factor.
The proposed method is so innovative that it has been applied for patenting. Thus, there are not many valuable references in literature, even the most recent.

Reviewer 2 Report
Comments and Suggestions for Authors
1. The title is precise and descriptive but could be enhanced to reflect the comparative or experimental nature of the work (e.g., "Comparative Study of Ultrasonic Atomization for Testing Liquid Metal Properties"); please clarify if this is a review of existing methods or includes new experimental insights.
2. Lines 10–21: The abstract is concise but misses critical technical specifics. Please Include details about the modeling parameters (like fluid properties and frequency range) and reference past studies briefly to contextualize the novelty of your approach.
3. Lines 25–41: This section explains the importance of material properties like viscosity and surface tension in metallurgical processes. While informative, it lacks recent citations. Include references from the last five years, especially related to ultrasonic atomization's role in characterizing these properties.
4. Lines 42–49: The explanation of viscosity and surface tension is basic. Can you provide equations or models related to these phenomena to align with the technical nature of the paper?
Lines 50–102 (Sections 2.1–2.3): The discussion on traditional methods (e.g., rotational viscometer, capillary method) is thorough. Also, Include a comparative table summarizing the pros/cons of each method with citations. This would improve clarity and highlight why ultrasonic atomization offers advantages.
5. Lines 103-131: The theoretical discussion about viscosity models is useful but dense. Can you please simplify the language for readability? “The generalized model by Budai et al. [9] effectively predicts alloy viscosities but struggles with incongruent melting alloys.
6. Lines 192–211 (Section 4): This section effectively introduces ultrasonic atomization. However, if you Add a citation for any study comparing ultrasonic and other atomization methods, it helps to understand (e.g., inert gas atomization).
7.Lines 221–234 (Equation 4): The derivation of droplet size based on capillary wavelength is insightful. Please Explain the practical significance of Equation 4. How does it compare to empirical observations or previous modeling studies?
7. Lines 286–308 (Section 6): The numerical model is clearly described. Can you add details on boundary conditions, assumptions, and limitations. Specify how mesh independence was validated for your results.
8. Lines 310–323: The simulation results align well with theoretical predictions. Please Explicitly compare your findings like droplet diameter of 36.3 μm) with experimental results from similar studies. This enhances the validity of your conclusions.
9. Lines 346–358 (Table 1): The table effectively summarizes material properties and outcomes. Can you please Add units and a brief note below the table explaining the significance of parameters like wave length and droplet diameter?
Lines 359–391: The discussion on droplet formation under varying amplitudes is detailed but could be more structured. Please Use subheadings (e.g., "Effect of Amplitude" and "Effect of Layer Thickness") for clarity.
10. Lines 359–391: The conclusions are supported by simulation results but lack experimental corroboration. Please Reiterate the importance of future experimental validation. Discuss potential industrial applications of your findings, such as in additive manufacturing or alloy development. References:
11. Lines 414–501: References are relevant but slightly dated; please Include more recent works on ultrasonic atomization techniques to strengthen the manuscript’s novelty
Comments on the Quality of English LanguageThe manuscript is technically dense and well-written but could benefit from more straightforward phrasing in some sections. For instance, replace “concerning the phenomenon of capillary waves” with “regarding capillary wave behavior.”
Author Response
Reply to Reviewer 2
Comments and Suggestions for Authors
First of all, we would like to thank you very much for your valuable comments on the article, they will help us in our future work. Below are our responses on specific comments.
- The title is precise and descriptive but could be enhanced to reflect the comparative or experimental nature of the work (e.g., "Comparative Study of Ultrasonic Atomization for Testing Liquid Metal Properties"); please clarify if this is a review of existing methods or includes new experimental insights.
The article contains new experimental observations and is a kind of introduction to the planned experiments In addition to a brief overview of existing methods. It contains too few examples and details to be considered a "comparative study". This decision was dictated by the desire to limit the length of the article and to present mainly ultrasonic atomization as a research method rather than as a detailed comparison.
- Lines 10–21: The abstract is concise but misses critical technical specifics. Please Include details about the modeling parameters (like fluid properties and frequency range) and reference past studies briefly to contextualize the novelty of your approach.
The information is more emphasized in the text. The properties of the analyzed fluids (liquid metals) are given in Table 1. The proposed method is innovative, has been applied for patent and there are no previous studies and results strictly in this subject.
- Lines 25–41: This section explains the importance of material properties like viscosity and surface tension in metallurgical processes. While informative, it lacks recent citations. Include references from the last five years, especially related to ultrasonic atomization's role in characterizing these properties.
For the present moment there are no valuable studies available on the investigation of the properties of liquid metals using ultrasonic atomization. The proposed method indicates ultrasonic atomization for the first time as a method to characterize the material properties of liquid metals.
- Lines 42–49: The explanation of viscosity and surface tension is basic. Can you provide equations or models related to these phenomena to align with the technical nature of the paper?
In the ultrasonic atomization section of the article, equations are given to relate the parameters of the atomization powder with the parameters of the atomization process itself and the properties of the atomized materials. The viscosity and surface tension models themselves were not developed in order not to over-expand the article and to fit within the character limit.
- Lines 50–102 (Sections 2.1–2.3): The discussion on traditional methods (e.g., rotational viscometer, capillary method) is thorough. Also, Include a comparative table summarizing the pros/cons of each method with citations. This would improve clarity and highlight why ultrasonic atomization offers advantages.
In the summary of the review of the most important existing methods, a paragraph was added that the above-mentioned methods are based on the study of the resistance of movement of a body immersed in liquid metal. Each method requires complex equipment and a relatively large amount of liquid metal. Ultrasonic atomization is not subject to these limitations.
- Lines 103-131: The theoretical discussion about viscosity models is useful but dense. Can you please simplify the language for readability? “The generalized model by Budai et al. [9] effectively predicts alloy viscosities but struggles with incongruent melting alloys.
The remark has been taken into account and text was modified. The text under consideration mainly refers to quotations from other available studies m.in. [3] and [9]. The authors did not want to oversimplify the text so that it would not lose the appropriate level of detail.
- Lines 192–211 (Section 4): This section effectively introduces ultrasonic atomization. However, if you Add a citation for any study comparing ultrasonic and other atomization methods, it helps to understand (e.g., inert gas atomization).
A reference has been added to the text.
- Lines 221–234 (Equation 4): The derivation of droplet size based on capillary wavelength is insightful. Please Explain the practical significance of Equation 4. How does it compare to empirical observations or previous modeling studies?
Appropriate wording has been added in the text.
- Lines 286–308 (Section 6): The numerical model is clearly described. Can you add details on boundary conditions, assumptions, and limitations. Specify how mesh independence was validated for your results.
The mesh parameters and boundary conditions were established to be consistent with the simulations described in the study by [42] (Sugondo et.al.) This makes it easier to verify the simulation results. Appropriate wording has been added in the text.
- Lines 310–323: The simulation results align well with theoretical predictions. Please Explicitly compare your findings like droplet diameter of 36.3 μm) with experimental results from similar studies. This enhances the validity of your conclusions.
An appropriate reference to similar results [24] and [28] has been added in the text.
- Lines 346–358 (Table 1): The table effectively summarizes material properties and outcomes. Can you please Add units and a brief note below the table explaining the significance of parameters like wave length and droplet diameter?
The entities are added in the table header. References to templates (3) and (4) have been added in the text. The diameter of the droplet depends on the wavelength in the liquid and on the properties of the atomized liquid. The wavelength itself in a liquid depends on the frequency of vibrations. More explanation was added to text.
- Lines 359–391: The discussion on droplet formation under varying amplitudes is detailed but could be more structured. Please Use subheadings (e.g., "Effect of Amplitude" and "Effect of Layer Thickness") for clarity.
The results shown on the formation of droplets under the influence of different amplitudes are only examples of results approximating some of the relationships occurring in the process and do not undertake a precise determination of the relationship between amplitude and layer thickness. These studies are planned at a further, experimental stage if it is necessary to determine such a relationship. This study, as a preliminary study, focuses on the static atomization of a uniform, spilled layer of liquid metal, while the actual atomization of liquid metal is a dynamic process and it is beneficial to atomize a portion of liquid metal as soon as possible, preferably before the spillage of a portion of liquid metal, in order to minimize heat transfer to the vibrating stack, especially the converter.
- Lines 359–391: The conclusions are supported by simulation results but lack experimental corroboration. Please Reiterate the importance of future experimental validation. Discuss potential industrial applications of your findings, such as in additive manufacturing or alloy development.
The research will continue, and a number of experiments are planned. The remark has been fully taken into account. Paragraph on future experimental validation, industrial applications and in the study of liquid metals, alloys, etc. is added.
- Lines 414–501: References are relevant but slightly dated; please Include more recent works on ultrasonic atomization techniques to strengthen the manuscript’s novelty.

Round 2
Reviewer 1 Report
Comments and Suggestions for Authors
Dear Authors,
Most of the suggestions made for the article were not applied in the article.
There is current literature that could be used, but it was discarded. In this situation, I regret to indicate that in my opinion the article is not fit for publication.
I wish you success in your research.
Sincerely,
Author Response
Dear Reviewer,
We provided corrections directly to the article text. Pls see changes marked green.
The reference you suggested (https://doi.org/10.3390/app13158984) is present in the article. Its results are invoked where possible compared and commented. Pls see References list number 28. Pls note that the ultrasonic system used in experiments was constructed by Mr Szostak-Staropietka..
Reference (https://www.mdpi.com/1996-1944/17/22/5642) contains some important information that wrongly adjusted parameters can make the process ineffective or impossible and it was included in references. This was a good suggestion and we thank you very much for it.
In the literature list, we have included all thematically related articles that we were able to find. However, we still cannot find results that can be directly compared with our results. We believe that this article will initiate a discussion on a new way to use the ultrasonic atomisation process.
Thank you very much for your valuable suggestions, which have been incorporated into the current version of the article, increasing its scientific value.
Best Regards,
Authors.
---------------------------------------------------------------------------------------------------------------
I would like to thank you for the opportunity to review the article. The research presents a good scientific contribution related to the atomization of liquid metal particles. However, adjustments are necessary for the publication of the article.
The following points need to be adjusted:
In the abstract, the knowledge gaps that the article intends to address are not clear. It is necessary that the objectives be indicated, as well as the steps of the research process, such as methodology, results, and conclusions, be briefly addressed.
In general, we have tried not to increase excessively the content of the introduction, where suggestions could be fully incorporated. Below are our detailed amendments.
Objective: To evaluate the feasibility of using ultrasonic atomization to study the properties of liquid metals.
Knowledge gaps: Searching for a new, possibly simple and cost-effective method to study the properties of liquid metals and clarifying the relationship between ultrasonic atomization parameters (amplitude, frequency, metal spill on vibrating surface) and obtained atomization results (average particle size, particle size distribution, atomization time)
Methodology: numerical simulation of the process as an introduction to a series of experiments.
Results: The simulation results are in line with the current state of knowledge about ultrasonic atomization. In the existing studies available to the authors, there is no data that can be compared directly.
Conclusion: Ultrasonic atomization can be applied to study the properties of liquid metals and this will be the subject of further research and experimentation.
In the introduction, the article presents a general approach under the aspects of reaction kinetics, viscosity, and surface tension. However, in a general approach in steelmaking, the condition of the slag, whether acidic or basic, the activities, fugacity, type of deoxidizers, among others, are fundamental in a macro analysis. It is recommended that only the metallurgy specifically used in the distributor used before atomization be addressed. At the end of the introduction, it is recommended to clearly state the objectives or gaps to which the research seeks answers, and these should be answered in the conclusions.
The proposed method is a method in the initial phase and at this stage it does not take into account, i.e the condition and amount of slag in the steel, but there are strong reasons to use this method also for testing slag or metal/alloy purity. At this moment these simulations were only for metals of high purity and homogeneity. Gaps: Finding the answer to the question whether ultrasonic atomization can be an effective tool to study the properties of liquid metals, including impurities. This will be examined later, in details.
The methodology contains information that can be used in the introduction. It is recommended that a flowchart be included to better understand the tests to be performed. The identification of different methods of testing liquid metal can be used in the introduction and not as a methodology. If it is only a review, it should be indicated in the abstract and title. Figure one should be better explained. The study was based on ferrous or non-ferrous alloys. Please clarify the focus of the material to be simulated, since the metallurgy involved has different nuances in each situation. Why does the article focus on gas atomization and not on water atomization?
The article mentions gas atomization as one of the methods of atomization, the only one that somehow related the results of the atomization process to the properties of atomized metals. For atomization with the use of other media (water, oil) there are no such studies. Water Atomization is a method very similar to atomization with gas, it is based on the same mechanisms as breaking a stream of liquid medium (metal) with another medium.
The figure caption has been supplemented with the phrase "an example of a complex and precise apparatus for measuring viscosity".
"Method flowchart" was added to in the "future work" paragraph:
- Computer simulation of the atomization process at the given parameters: frequency, amplitude, thickness of the atomized liquid layer, properties of the atomized material.
- Theoretical determination of the minimum amplitude and the influence of the shape and thickness of the atomized liquid layer on the time of the atomization process.
- Conducting atomization experiments for specific process parameters (specific frequency, specific amplitude).
- Investigation of the influence (separately) of the amplitude, frequency, shape and thickness of the liquid metal layer) and the parameters of the atomized liquid (density, viscosity, surface tension, purity) on the time of the atomization process and the size of particles.
- The results require further discussion with other authors and correlations of the simulation using ultrasound atomization with other techniques.
This statement has been added to the body of the article. Refeences to similar results [24] and [28] are added with the note that this is a slightly different method of atomization.
- There are results in the conclusions. Ideally, the conclusions should respond to the knowledge areas identified at the end of the introduction.
A paragraph has been added to the text. The results of the simulation described in the article refer to one specific case of atomization, at a specific frequency and amplitude. These results are consistent with the available theoretical results and results described in [24] and [28]. There are no other published results of the atomization process with parameters as in the article that could be directly compared.
- The references should be updated and have a higher impact factor.
The proposed method is so innovative that it has been applied for patenting. Thus, there are not many valuable references in literature, even the most recent.

Round 3
Reviewer 1 Report
Comments and Suggestions for Authors
Dear Authors,
In this second review, the article was modified to meet almost all of the suggestions for improvements. In this way, it is ready for publication and contains important data.
Regarding the discussion of results, even though the authors did not find current articles specifically on the topic, there is information on some of the topics presented that is very current and relevant.
Sincerely,